# Nano-Sized Chimeric Human Papillomavirus-16 L1 Virus-like Particles Displaying *Mycobacterium tuberculosis* Antigen Ag85B Enhance Ag85B-Specific Immune Responses in Female C57BL/c Mice

**DOI:** 10.3390/v15102123

**Published:** 2023-10-19

**Authors:** Fangbin Zhou, Dongmei Zhang

**Affiliations:** Department of Tropical Diseases, Naval Medical University, Shanghai 200433, China

**Keywords:** virus-like particle, Ag85B, tuberculosis, *Mycobacterium tuberculosis*, vaccine

## Abstract

Bacillus Calmette–Guerin (BCG), the only current vaccine against tuberculosis (TB) that is licensed in clinics, successfully protects infants and young children against several TB types, such as TB meningitis and miliary TB, but it is ineffective in protecting adolescents and adults against pulmonary TB. Thus, it is a matter of the utmost urgency to develop an improved and efficient TB vaccine. In this milieu, virus-like particles (VLPs) exhibit excellent characteristics in the field of vaccine development due to their numerous characteristics, including but not limited to their good safety without the risk of infection, their ability to mimic the size and structure of original viruses, and their ability to display foreign antigens on their surface to enhance the immune response. In this study, the HPV16 L1 capsid protein (HPV16L1) acted as a structural vaccine scaffold, and the extracellular domain of Ag85B was selected as the *M. tb* immunogen and inserted into the FG loop of the HPV16 L1 protein to construct chimeric HPV16L1/Ag85B VLPs. The chimeric HPV16L1/Ag85B VLPs were produced via the *Pichia pastoris* expression system and purified via discontinuous Optiprep density gradient centrifugation. The humoral and T cell-mediated immune response induced by the chimeric HPV16L1/Ag85B VLP was studied in female C57BL/c mice. We demonstrated that the insertion of the extracellular domain of Ag85B into the FG loop of HPV16L1 did not affect the in vitro stability and self-assembly of the chimeric HPV16L1/Ag85B VLPs. Importantly, it did not interfere with the immunogenicity of Ag85B. We observed that the chimeric HPV16L1/Ag85B VLPs induced higher Ag85B-specific antibody responses and elicited significant Ag85B-specific T cell immune responses in female C57BL/c mice compared with recombinant Ag85B. Our findings provide new insights into the development of novel chimeric HPV16L1/TB VLP-based vaccine platforms for controlling TB infection, which are urgently required in low-income and developing countries.

## 1. Introduction

Tuberculosis (TB), an ancient infectious disease caused by *Mycobacterium tuberculosis* (*M. tb*), remains a threat to the lives of several million people around the world. According to the World Health Organization (WHO), there were about 10.6 million cases infected with TB and 1.6 million deaths from this disease in 2021, including 0.19 million concurrent infections with HIV [1]. Besides the side effect of the coronavirus (COVID-19) pandemic that temporarily interrupted the annual decline in TB cases [2], the potential threat from people with a latent tuberculosis infection (LTBI), the increase in the number of cases of concurrent infection with HIV, and the spread of multidrug-resistant tuberculosis (MDR-TB) and extensively drug-resistant tuberculosis (XDR-TB) also contribute to the difficulty in effectively controlling TB. More importantly, vaccination is inevitable to achieve the goal of ending TB; however, the only licensed vaccine, Bacillus Calmette–Guerin (BCG), is limited in preventing TB due to its highly variable efficacy [3]. It has been reported that BCG can successfully protect infants and young children against TB meningitis and miliary TB, but it is ineffective in protecting adolescents and adults against pulmonary TB, which is the most prevalent form of TB [4]. Therefore, novel TB vaccines with a low cost and high efficiency are urgently required.

Strategies to improve TB vaccines rely heavily on the identification of novel TB antigens involved in protective immunity. Among a dozen of M. tb immunogens identified, members of the antigen 85 (Ag85) complex (Ag85A, Ag85B, and Ag85C) play essential and vital roles in the virulence of M. tb via binding to fibronectin and elastin proteins, mycolyltransferase activity, as well as through phagosome maturation prevention in macrophages, and are being utilized as tools in the construction of new vaccines [5]. At present, there are 14 TB vaccine candidates in clinical trials, which can be categorized into live attenuated mycobacterial vaccines, killed mycobacterial vaccines, adjuvanted subunit vaccines, and viral vector vaccines [6]. Of these, adjuvanted subunit and viral vector vaccines are the most abundant types of TB vaccines in clinical evaluation [7,8,9,10,11], and most of them use members of the Ag85 complex as key antigens. For example, compared with the earlier H1:IC31 vaccine (Ag85B-ESAT6 fusion protein), the latency antigen, Rv2660c, was added to H56:IC31. A preclinical test indicated that H56 confers protective immunity characterized by a more efficient containment of late-stage infection than the Ag85B-ESAT6 vaccine (H1) and BCG [12]. GamTBVac uses a novel vaccine formulation, in which three M. tb antigens, namely Ag85A, ESAT6, and CFP-10, are fused with a dextran-binding domain and formulated with an adjuvant (TLR9 agonist) consisting of a Dextran 500 kDa and DEAE-Dextran 500 kDa core covered with CpG oligonucleotides [13]. In murine and guinea pig TB models, GamTBvac had a strong immunogenicity and showed a particularly strong protective effect as a BCG booster vaccine [13]. Viral vectored vaccines, including AdHu5Ag85A, TB/Flu-04L, and ChAdOx1.85A+MVA85A, also use recombinant attenuated influenza or type 5 human adenovirus vector to express Ag85 [14,15,16]. Moreover, it is noted that multiple vaccine candidates constructed with Ag85 are currently being developed in preclinical animal model phases [17,18]. BCG85C5, a recombinant BCG overexpressing Ag85A and the C5 peptide of CFP-10, reduced bacterial loads 30 days after the initial infection and 30 days after re-infection in mice compared to BCG [19]. rBCG:CysVac2, a recombinant BCG expressing CysVac2, a fusion protein of Ag85A and CysD, showed similar bacterial burdens in the lungs both 4 and 12 weeks post infection compared to BCG in mice [20]. SeV85AB, a recombinant SeV-vectored vaccine, uses an SeV vector to express Ag85A and Ag85B. Compared with BCG, SeV85AB vaccination can achieve a higher level of TRM-mediated immune response in mucosal tissues to protect against M. tb invasion in the early phase of infection [21]. However, there are no vaccines with Ag85 that are licensed for use that are superior to BCG no matter how many strategies are improved based on the optimization of the current BCG vaccine or the development of novel vaccines such as subunit and vectored vaccines.

Another strategy to develop TB vaccines is the improvement of vaccine construction methods. Virus-like particles (VLPs) are hollow particles that contain one or more self-assembled structural proteins of a virus but do not contain viral nucleic acids, commonly known as pseudoviral particles [22]. As a new type of subunit vaccine, VLPs present excellent prospects in the field of vaccine development and have several advantages as follows. Firstly, compared with a single protein or peptide, the conformational epitopes of VLPs are more similar to those of original viruses, thus significantly enhancing the level of immune response. Secondly, without affecting the structure of VLPs, some targeted amino acid sequences can be inserted or deleted in accordance with the requirements, and artificial modifications can be made to construct chimeric VLPs. In addition, VLPs can also be developed as carriers to deliver some small molecules or drugs, or as carriers for the delivery of DNA or RNA vaccines to improve their immune efficacy or for gene therapy [23]. Since the discovery in the 1980s that the capsule proteins of polyoma viruses can self-assemble into VLPs, the research on VLP vaccines has been rapidly developed. To date, several VLP-based vaccines, including vaccines against human papilloma virus (HPV), hepatitis B virus (HBV), and malaria, have been successfully licensed for use [24,25]. Additionally, human VLP vaccines for influenza virus (IV), HIV, and Ebola virus (EBoV) are also under development.

In this study, HPV-16 L1 VLPs were used as carriers to insert the *M. tb* antigen, Ag85B, and construct the chimeric molecule of HPV-16 L1/Ag85B. Western blotting (WB) confirmed the successful expression of the chimeric HPV16L1/Ag85B in *Pichia pastoris*. The electron transmission microscopy (TEM) examination results showed that chimeric HPV-16 L1/Ag85B could form particles with a diameter of approximately 50 nm, and the immunogenicity of HPV-16 L1/Ag85B was also evaluated.

## 2. Methods and Materials

### 2.1. Cloning and Expression of Recombinant Ag85B in E. coli

The gene of the *M. tb* antigen, Ag85B, was amplified via PCR using the *M. tb* H37Rv strain’s genomic DNA as a template. The purified PCR product was then cloned into the *pET-28a* expression vector, and the recombinant antigen was expressed with a C-terminal His-tag. Recombinant antigen expression in *Escherichia coli* (*E. coli*) and *Rosetta* (DE3; Novagen, Germany) was induced with 1 mM isopropyl-β-D-thiogalactoside (IPTG) and analyzed via WB. The inclusion bodies were denatured via the addition of 0.05 mM TCEP and 4.4% N-lauroylsarcosine and refolded in a universal refolding buffer C7 (1.0 mM TCEP, 250 mM NaCl, 12.5 mM β-cyclodextrin, 0.5 M L-arginine, 50 mM Tris-HCl, pH 7.5) from the iFOLD Protein Refolding System 1, as described previously [26]. A nickel column was used to purify recombinant Ag85B via affinity chromatography, and purified recombinant proteins were examined vis sodium dodecyl sulfate polyacrylamide gel electrophoresis (SDS-PAGE) and quantified using the Beyotime BCA protein quantitation kits. The purity was determined using Quality One software.

### 2.2. Modification of HPV16L1

The FG loop of HPV16L1 was deleted, and HPV16L1 was modified as follows: The sequence on the left side of the FG loop was first amplified using the paired primers, F1 and R1 (Table 1), which were designed on the N-terminal of HPV-16L1 and the left side of the FG loop sequence with two added restriction sites, BamHI and NcoI. The sequence on the right side of the FG loop was then amplified using the paired primers, F2 and R2, designed on the C-terminal of HPV-16L1 and the right side of the FG loop sequence with two added restriction sites, EcoRI and NcoI. The above left sequence was cloned into the pBluscript Ⅱ SK + (pBlu) plasmid using BamHI and NcoI, and the right sequence was then inserted into the recombinant pBlu vector carrying the left sequence using NcoI and EcoRI, thus obtaining the HPV-16 L1 gene without the FG loop and forming a restriction site, NcoI.

### 2.3. Construction of Chimeric HPV16L1/Ag85B 

The transmembrane domain of Ag85B was predicted using TMHMM software (http://www.cbs.dtu.dk/services/TMHMM/, accessed on 1 October 2023) [27], and the codon usage of the extracellular domain identified between amino acid positions 41 and 285 was adapted to the bias of *Pichia pastoris*. An initiating ATG codon in a Kozak consensus sequence was required and inserted for proper translation initiation of the gene. The same restriction site, NcoI, was added to both ends of the optimized sequence via PCR. The above fragment was then ligated to the recombinant pBlu vector carrying the modified HPV16L1 with the same restriction, which needed to be dephosphorylated before. The HPV16L1-Ag85B fusion sequence was digested from two restriction sites, BamHI and EcoRI, of the recombinant pBlu vector and cloned into the *Pichia* expression vector, pPIC3.5K, generating a recombinant pPIC3.5K plasmid construct. The recombinant pPIC3.5K construct was transformed via electroporation into competent *E. coli* TOP10 cells, and ampicillin-resistant colonies were selected. Plasmid DNA was isolated using a miniprep for sequencing to confirm that ATG was in the proper context for eukaryotic translation initiation. The 5′ and 3′ AOX1 sequencing primers (F3 and R3) were used to sequence the constructs.

### 2.4. Expression and Purification of HPV16L1/Ag85B VLP in Pichia Pastoris

A sufficient amount of recombinant pPIC3.5K plasmid DNA was prepared and linearized using the restriction enzyme SalI for insertion at *HIS4* prior to transformation into *Pichia*. The linearized construct in pPIC3.5K was transformed into GS115 via electroporation, generating both Mut^+^ and Mut^S^ recombinants. The parent vector linearized in the same manner as the construct was used as a control to confirm the integration via PCR and was used as a control for background for the expression analysis and the quantitative dot blots or Southern blot analysis. Another control was pPIC3.5K containing one copy of the expression cassette. His+ transformants were generated using recombinant pPIC3.5K and screened in vivo to obtain 10 Geneticin^®^-resistant colonies to test. The colonies with multiple copies were selected to proceed directly to overexpressing the proteins. Cells were harvested after 96 h of incubation at 37° and ultracentrifuged at 136,000× *g* at 4 °C for 10 min. Pellets resuspended in sorbitol solution with a pH of 5.8–7.2 were lysed with snailase (10 mg/mL) at 37 °C for 1 h, and the supernatants containing HPV16L1/Ag85B VLPs were collected at 136,000× *g* at 4 °C for 10 min. VLPs were further purified via discontinuous Optiprep density gradient centrifugation (27–33–39%) at 234,000× *g* at 4 °C for 4 h and detected using SDS-PAGE. 

### 2.5. Negative Staining and Transmission Electron Microscopy (TEM)

HVP16L1 and HVP16L1/Ag85B VLPs equilibrated with 20 mM Tris-HCl (pH 7.4; 137 mM NaCl) were layered onto copper TEM grids for 1 min, and negatively stained using 2% uranyl acetate solution for 1 min. The grids were placed in a dehumidifier chamber at least 2 h before observation. Images were acquired using a TEM at a magnification of 150,000.

### 2.6. Immunization of Mice, Collection of Sera, and Isolation of Splenocytes

Six- to eight-week-old C57BL/c mice were used in this study, and all experiments were approved by the Institutional Animal Ethics Committee of Navy Medical University. The mice were immunized by priming them with BCG, followed by boosting with vaccine candidates twice at two-week intervals. Purified recombinant Ag85B, HPV16L1, and HPV16L1/Ag85B VLPs were emulsified with a moderate volume of Freund’s incomplete adjuvant and PBS. In the experimental groups, the mice were primed with 10^6^ CFU of BCG intradermally (i.d) and boosted with 20 µg of recombinant Ag85B and HPV16L1/Ag85B VLPs intramuscularly (i.m). In the control groups, the mice were primed with BCG followed by immunization with HPV16L1 VLPs and PBS. Blood samples were collected before priming with BCG and after one or two weeks per immunization and stored at −80 °C for the antibody assay. The mice were sacrificed, and spleen samples were collected two weeks after the final immunization for cytokine analysis. The spleens were crushed and pressed individually through a 70 µm cell strainer (BD Falcon, Bedford, MA, USA) with a 5 mL syringe rubber plunger. The red blood cells were lysed using ammonium chloride–potassium (ACK) lysis buffer (150 mMNH_4_Cl, 10 mM KHCO_3_, 0.1 mM EDTA, pH 7.4) and then resuspended in warm RPMI-1640 (Sigma-Aldrich, St. Louis, MD, USA).

### 2.7. Evaluation of Humoral Immune Response

An enzyme-linked immunosorbent assay (ELISA) was used for the detection of anti-Ag85B antibodies in the sera of the immunized animals. Briefly, the purified protein derivative (PPD) or the Ag85B protein was adjusted to 2 μg/mL with coating buffer (0.05 M Na_2_CO_3_-NaHCO_3_, pH 9.6), and 100 µL was added to each well of 96-well Immunosorp plates (Nunc, Weston, FL, USA) for incubation overnight at 4 °C. The plates were washed three times with 375 μL PBST (137 mM NaCl, 2.7 mM KCl, 10 mM Na_2_HPO_4_, 2 mM KH_2_PO_4_, and 0.05% Tween-20) and then blocked in 0.2 mL 3% bovine serum albumin in PBS for 2 h at room temperature. After three washes, the collected serum samples (diluted 1:200 in blocking buffer) were added to each well and incubated at 37 °C for 1 h. After five washes five times, 100 µL of 1:5000-diluted HRP-conjugated anti-human IgG, IgG1, or IgG2 secondary antibody (Promega, Madison, WI, USA) was added for incubation at 37 °C for a further 1 h. After five washes, the color reaction was developed with 100 µL TMB (3, 3′, 5, 5′-tetramethylbenzidine) substrate solution (eBioscience, San Diego, CA, USA) and stopped via the addition of 50 µL 2 N H2SO4 (eBioscience, San Diego, CA, USA). The optical density (OD) was measured and recorded at a wavelength of 450 nm using a microplate reader (ELX50, Bio-Tek Instruments, Salem, MA, USA).

### 2.8. Cytokine Analysis

Splenocytes from the immunized mice were cultured at a density of 1 × 10^6^ cells/well in 96-well plates (BD Biosciences, San Jose, CA, USA ) in RPMI 1640 supplemented with 1% penicillin-streptomycin, 1 mM glutamine, and 10% fetal bovine serum (FBS) at 37 °C with 5% CO_2_. Triplicate wells were stimulated with GST (negative control; Abcam, Cambridge, MA, USA), LEAF-CD3ε (positive control; Biolegend, San Diego, CA, USA), PPD (Chengdu Institute of Biological Products Co., Ltd., Chengdu, China), and Ag85B antigen (Abcam, Cambridge, MA, USA). Supernatants were collected after 48 h and IFN-γ, interleukin (IL)-2, tumor necrosis factor (TNF), and IL-10 were quantified using a cytokine ELISA (eBioscience, San Diego, CA, USA).

### 2.9. Statistical Analysis

Statistical analysis was performed using Prism GraphPad software version 8.01 (San Diego, CA, USA). Statistical analysis was carried out through parametric testing of mean values via one-way analysis of variance (with Tukey’s post hoc testing to identify between-group significance).

## 3. Results

### 3.1. Characterization of Recombinant M. tb Antigen Ag85B

The workflow for producing recombinant Ag85B in *E. coli* is shown in Figure 1A. Briefly, the gene sequence of the *M. tb* antigen, Ag85B, amplified via PCR from the H37Rv strain’s genomic DNA template, was cloned into a pET-28a expression plasmid (Figure 2A) and expressed using the *E. coli* expression system with a C-terminal 6 × His-tag. SDS-PAGE and WB indicated that recombinant Ag85B mainly existed in the form of an insoluble inclusion body, but not the culture supernatant (Figure 2B,C). A nickel column was used to purify recombinant Ag85B via affinity chromatography, and the purity calculated using the Quality One software version 4.6 (Bio-rad, Irvine, CA, USA) was 91.66%. The yield of recombinant Ag85B was about 2.5 mg/100 mL 2 × YT liquid medium.

### 3.2. Modification of HPV16L1 and Construction of HPV16L1/Ag85B

HPV16L1 contains 505 amino acids in its full length and is capable of self-assembly into single-layer L1 VLPs; it is also characterized by three virus-facing hinge regions between beta folds: DE, FG, and HI loops (Figure 3A) [28]. Previous research showed that the insertion of moderate-length foreign sequences into the FG loop did not influence the formation of VLPs. In the current study, a modified HPV16L1 was constructed, in which the FG loop was deleted, and a restriction site, NcoI, was added accordingly (Figure 3B). The sequence of the extracellular domain of the *M.tb* antigen Ag85B was inserted into the modified HPV16L1, which acted as a structural vaccine scaffold and led to the construction of a chimeric HPV16L1/Ag85B protein. The monomer structures of HPV16 L1 and modified HPV16 and HPV16L1/Ag85B were preliminarily predicted using the SWISS-model server. Firstly, the full sequence of HPV16L1 was uploaded at the server to search and identify suitable templates, and the result indicated that 7CN2 is a PDB entry of HPV16L1. 7cn2.1.R with the highest identify (99.80%) was then chosen as the structural template to build the HPV16 L1 and modified HPV16 and HPV16L1/Ag85B homology models. Compared with the conformation of HPV16L1, the FG loop was successfully removed with the modified HPV16 and correspondingly replaced by the extracellular domain of Ag85B in the chimeric HPV16L1/Ag85B (Figure 3C). Since HPV16L1 could homogeneously self-assemble into T = 7 icosahedral particles with 72 pentameric capsomeres [29], the high-density display of Ag85B on the exterior surface of chimeric HPV16L1/Ag85B VLPs is potentially highly immunostimulatory and could lead to the induction of epitope-specific immune responses.

### 3.3. Production of HPV16L1/Ag85B VLPs Using Pichia Pastoris Expression System

The workflow for producing the chimeric HPV16L1/Ag85B VLPs in *Pichia pastoris* is shown in Figure 1B. Briefly, the optimized codon usage of the extracellular domain of Ag85B was first inserted into the recombinant pBlu vector carrying the modified HPV16L1. The HPV16L1-Ag85B fusion sequence (2124 nucleotide length) digested by the recombinant pBlu vector was then cloned into the *Pichia* expression vector, pPIC3.5K, generating a recombinant pPIC3.5K plasmid construct (Figure 4A). The construct was further transferred into *Pichia pastoris* via electroporation and the colonies with multiple copies were selected to proceed directly to producing the proteins. Anti-Ag85B polyclonal antibody was selected to recognize Ag85B epitope, and anti-HPV16 monoclonal antibody was used to recognize HPV16L1. WB indicated that the chimeric HPV16L1/Ag85B was successfully expressed by the *Pichia pastoris* expression system (Figure 4B).

The produced HPV16L1/Ag85B was purified via discontinuous Optiprep density gradient centrifugation (27–33–39%) at 234,000× *g* at 4 °C for 4 h and detected via SDS-PAGE and WB using the anti-HPV16L1 antibody (Figure 4C,D). TEM was further used to investigate the morphology of the HPV16L1/Ag85B VLPs. VLPs with a diameter of approximately 50 nm were observed (Figure 5), and it seemed that the insertion of a moderate-length Ag85B sequence into the FG loop did not significantly influence the formation of HPV16L1 VLPs.

### 3.4. HPV16L1/Ag85B VLP Immunization Induced Antigen-Specific Antibody Responses

Twenty female C57BL/c mice were immunized by priming them with BCG, followed by boosting with vaccine candidates twice at two-week intervals, as shown in Figure 6A. To investigate whether an Ag85B-specific antibody response could be induced by HPV16L1/Ag85B VLP immunization, serum samples from the different immunized groups at different vaccination time points were collected, and the antibody titers of IgG, and two of its subtypes, IgG1 and IgG2a, were detected via an indirect ELISA coated with recombinant Ag85B. It was observed that the IgG antibodies were not significantly increased among all groups after seven weeks of priming with BCG (Figure 6B). However, a statistically significant difference in the Ag85B-specific IgG was detected in the recombinant Ag85B and HPV16L1/Ag85B VLP groups in comparison with the PBS and HPV16L1 control groups. Compared to the recombinant Ag85B group, the HPV16L1/Ag85B VLP group elicited a higher anti-Ag85B-specific IgG antibody titer at nine (*p* = 0.0004) and twelve weeks (*p* = 0.0002). Moreover, it was found that both recombinant Ag85B and HPV16L1/Ag85B VLPs mainly enhanced the anti-Ag85B IgG2a titer, but not the IgG1 titer, which led to a significant increase in the anti-Ag85B IgG2a/IgG1 ratios (Figure 6C,D) and suggested that the immune response induced by these proteins in the mice was biased towards the Th1 type.

### 3.5. HPV16L1/Ag85B VLP Immunization Induced Significantly Higher Ag85B-Specific IFN-γ + T Cell Responses

The above results support the high potency of HPV16L1/Ag85B VLPs as immunogens to elicit potent humoral immune responses against the surface-displayed vaccine antigen, Ag85B. Next, we explored the potency of HPV16L1/Ag85B VLPs to elicit T cell immune responses against the surface-displayed vaccine antigen, Ag85B. The separated splenocytes from the four groups were all stimulated with GST, CD3ε antibody, PPD, and Ag85B, and the supernatants were collected after 48 h to be used for IFN-γ, interleukin (IL)-2, tumor necrosis factor (TNF)-α, and IL-10 quantification. As shown in Figure 7A, IFN-γ secretion was almost undetectable among all groups after splenocyte stimulation with GST, which is an irrelevant *schistosoma japonica* recombinant protein that was set as a negative control. Conversely, as expected, high levels of IFN-γ were observed among all groups when stimulated with the LEAF-CD3ε antibody, which was set as a positive control. There was no difference between the HPV16 L1 and PBS groups regarding IFN-γ secretion after splenocyte stimulation with PPD. However, recombinant Ag85B induced a higher frequency of IFN-γ secretion compared with that of HPV16 L1. As compared with recombinant Ag85B, HPV16L1/Ag85B VLPs showed a significant difference in IFN-γ secretion (*p* = 0.028). Regarding the Ag85B-specific T cell responses, the highest concentration of IFN-γ secretion/10^6^ splenocytes was observed in the HPV16L1/Ag85B VLP immunized mice compared to the mice that received PBS, HPV16L1, or recombinant Ag85B (Figure 7A). A significantly higher IFN-γ secretion was observed in the mice that were vaccinated with recombinant Ag85B VLPs in comparison with the mice that received HPV16L1. In addition, as compared to recombinant Ag85B, the HPV16L1/Ag85B VLPs elicited a higher anti-Ag85B IFN-γ level, although the difference did not reach a statistically significant level (*p* = 0.156). IL-2, TNF-α, and IL-10 were also quantified among all groups; however, there were no significant differences between them whether the splenocytes were stimulated with PPD or Ag85B (Figure 7B–D).

## 4. Discussion

TB is currently the focus of many vaccine studies as a result of the inability of BCG to end TB. Although a dozen TB vaccine candidates have been developed in clinical stages, novel vaccine construction approaches still deserve to be explored. In this study, we demonstrated that (1) the *Pichia pastoris* expression system and the discontinuous Optiprep density gradient centrifugation method could be feasible approaches to produce and purify chimeric HPV16L1/Ag85B VLPs; (2) HPV 16 L1 could act as a structural vaccine scaffold, and the insertion of the extracellular domain of Ag85B into the FG loop of HPV16 L1 did not affect the in vitro stability and self-assembly of chimeric HPV16L1/Ag85B VLPs; (3) the sequential Ag85B peptide exposed to the FG loop of chimeric HPV16L1/Ag85B VLPs could be detected by an anti-Ag85B antibody in vitro; and (4) chimeric HPV16L1/Ag85B VLPs could induce Ag85B-specific antibody responses and elicit Ag85B-specific T cell responses in mice. These findings contribute to the development of novel chimeric HPV16L1/Ag85B VLP-based vaccine platforms for controlling TB infection, which are urgently needed in developing and industrialized countries.

Several expression platforms, including eukaryotic, prokaryotic, and cell-free systems derived from eukaryotic or prokaryotic expression systems, can be employed for producing VLPs. About 70% of reported VLPs are prepared with eukaryotic systems, which include yeast systems, mammalian cell systems, baculovirus/insect cell (B/IC) systems, and plant systems [30]. *Pichia pastoris*, *Saccharomyces cerevisiae*, and *Saccharomyces polymorphus* are the most widely used yeast cells, which can be used to carry out the post-translational modification (PTM) of the expressed proteins to ensure they have the correct conformations and biological activities. In this study, *Pichia pastoris* was employed to successfully express HPV16L1/Ag85B VLPs, and the products were then purified via the discontinuous Optiprep density gradient centrifugation method. Although chromatographic purification is a common method used to purify VLP proteins, we used gradient centrifugation to achieve a similar purification effect in this study.

HPV16L1 contains 505 amino acids in its full length and is capable of self-assembly into VLPs; it is also characterized by three virus-facing hinge regions between beta folds: DE, FG, and HI loops. Numerous studies have shown that the insertion of moderate-length foreign sequences into these loops did not influence the formation of VLPs [31,32,33]. Moreover, the inserted foreign sequences located within these surface-exposed loops could be successfully presented in chimeric HPV16L1/Ag85B VLPs. In this study, the FG loop was chosen as the preliminary insertion region of HPV16L1 since its sequence length is larger than that of the DE and HI loops, and thus, it was theoretically possible to insert larger fragments of Ag85B without affecting the formation of the VLPs. The results also indicated that the extracellular domain of Ag85B insertion did not modify the HPV16L1 VLP structure, and the chimeric HPV16L1/Ag85B protein still had the capacity to form VLPs. Moreover, the inserted Ag85B sequences located within the surface-exposed FG loop could be recognized by the anti-Ag85B polyclonal antibody, demonstrating their proper presentation on the chimeric HPV16L1/Ag85B VLPs.

In the present study, the augmentation of antigen-specific immune responses via HPV16L1/Ag85B VLPs led to a higher induction of antigen-specific IgG and IgG2a antibodies, and accumulated IFN-γ secretion was also observed in the HPV16L1/Ag85B VLP vaccination group compared with the recombinant Ag85B group, indicating a biased Th1 type induced by the HPV16L1/Ag85B VLPs. However, it is worth noting that the parameters for evaluating an eligible vaccine are too simple, and too much attention is paid to the Th1 immune response effect induced by vaccines. A review study showed that six TB vaccine candidates, namely MVA85A, AERAS-402, H1:IC31, H56:IC31, M72/AS01E, and ID93:GLA-SE, evoked highly similar functional properties of memory T cell responses, indicating a lack of diversity among the available TB vaccine candidates [34]. Although an effective Th1-type cell-mediated adaptive immune response characterized by the secretion of IFN-γ and TNF via antigen-specific CD4 T cells is known to be required, it is not sufficient to provide protective immunity against TB. For example, numerous studies indicated that humoral immunity also plays an important role in anti-TB immunity [35,36,37]. In view of the complexity of TB–host interactions, the mechanism related to the immune response to TB is still poorly understood. If the evaluation index of a qualified vaccine is too simple, it will greatly increase the risk of vaccine failure; thus, new biological markers of vaccine protective immunity are urgently needed.

Most vaccines employ a limited pool of immunologically dominant target antigens, mainly from the secretion of Ag85 and ESAT-6 protein families. In mice, it has been observed that the limitation of antigen availability weakens the Ag85B-induced protective immunity during chronic infection, while the immunity of ESAT-6-specific T cells is restricted as a result of functional exhaustion, highlighting potential challenges in employing these antigens for vaccine development [38]. As the antigen-specific natural T cell response of *M. tb* is highly heterogeneous, tens of antigens are required to cover 80% of CD4 T cell responses. However, current vaccine approaches utilizing a few antigens may not induce a sufficient immune response. Conversely, antigens that are poorly recognized during natural infection (referred to as unnatural antigens) may not be fully recognized by the immune system at all during infection, and their role in protection is unclear and worthy of further investigation. The subunit vaccine, M72: AS01E, is composed of unnatural antigens, Rv0125 and Rv1196, combined with the adjuvant AS01. Phase 1 and 2 clinical trials in *M. tb*-negative populations have shown that M72:AS01E has a relatively high T cell immune induction effect [11]. In this study, Ag85B was employed to construct chemic HPV16L1/TB VLPs, making it feasible to develop a novel chimeric HPV16L1/TB VLP-based vaccine platform to control TB infection. Several novel TB vaccine candidates have been identified by our research group, and evaluations of the immunogenicity and immunocompetence of these chimeric HPV16L1/TB VLPs are ongoing.

All in all, VLP-based vaccines have gradually become a hotspot in the field of vaccine development due to their numerous characteristics, including but not limited to their good safety without a risk of infection and their abilities to mimic the size and structure of original viruses and to display foreign antigens on their surface to enhance the immune response. To date, some VLP-based vaccines, including those for IV, HBV, HEV, HPV, and malaria, have been successfully licensed in the market [22], and VLP-based vaccines designed for HIV, EBoV, and other infectious diseases are in preclinical development or clinical trials [39]. Some advances in the development of VLP-based TB vaccines have also been made [40,41,42,43], the key to which lies in finding novel antigens with high immunogenicity and improving the form of vaccine construction to stimulate the immune system. It is believed that novel TB antigenic targets and VLP carriers will be continuously identified and exploited through in-depth research on the relationship between TB and host immunity, and more safe and effective vaccines must be developed to prevent TB in humans in the future.

## Figures and Tables

**Figure 1 viruses-15-02123-f001:**
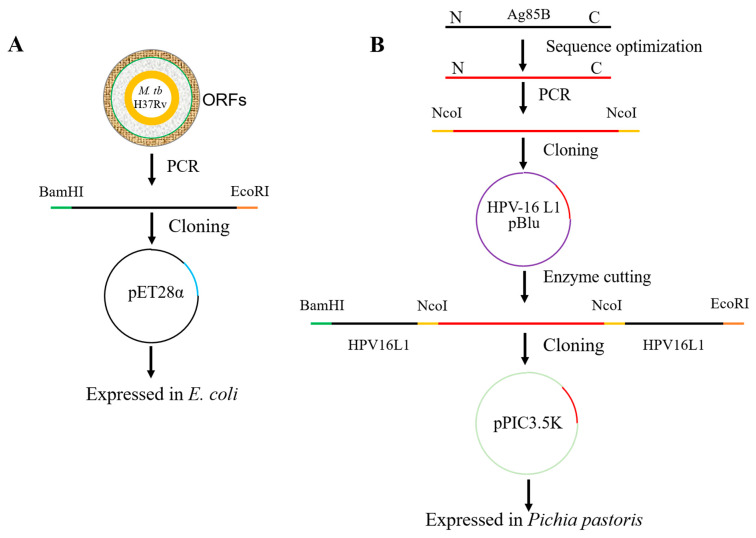
The workflow of expression of *M. tb* antigens. (**A**) The workflow of producing recombinant Ag85B in *E. coli.* The sequence of Ag85B was amplified via PCR using the *M. tb* H37Rv strain genomic DNA as a template. The purified PCR product was then cloned into *pET-28a* expression vector. The recombinant antigen was expressed in *E. coli.* (**B**) The workflow of producing the chimeric HPV16L1/Ag85B VLPs in *Pichia pastoris.* The codon usage of the extracellular domain of Ag85B was adapted to the bias of *Pichia pastoris*. Both ends of the optimized sequence were added to the same restriction site NcoI via PCR. The above fragment was then ligated into the recombinant pBlu vector carrying the modified HPV16L1 with the same restriction. The fusion HPV16L1-Ag85B sequence was digested from two restriction sites, BamHI and EcoRI, of the recombinant pBlu vector and cloned into the Pichia expression vector, pPIC3.5K, generating a recombinant pPIC3.5K plasmid construction. The chimeric HPV16L1/Ag85B VLPs were expressed in *Pichia pastoris*.

**Figure 2 viruses-15-02123-f002:**
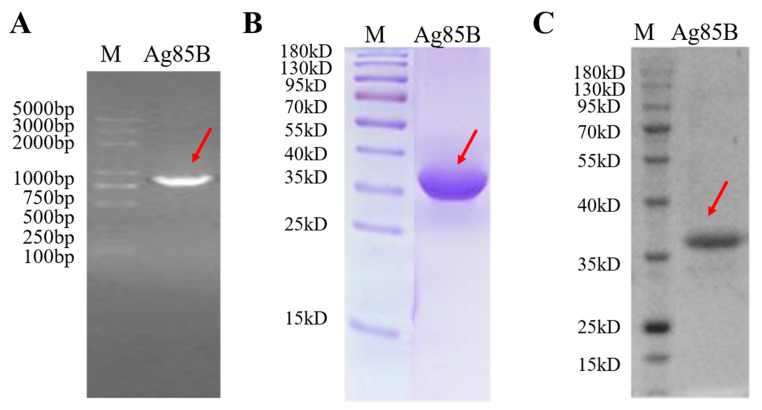
Production of recombinant Ag85B in *E. coli*. The red arrow indicates the PCR product. (**A**) The gene sequence of the *M. tb* antigen, Ag85B, was amplified via PCR. (**B**,**C**) SDS-PAGE and WB analysis of purified recombinant Ag85B. The recombinant Ag85B was expressed in *E. coli* and was induced with 1 mM IPTG. Purified recombinant proteins were examined via SDS-PAGE and WB. The red arrow indicates the position of recombinant Ag85B (37 kDa).

**Figure 3 viruses-15-02123-f003:**
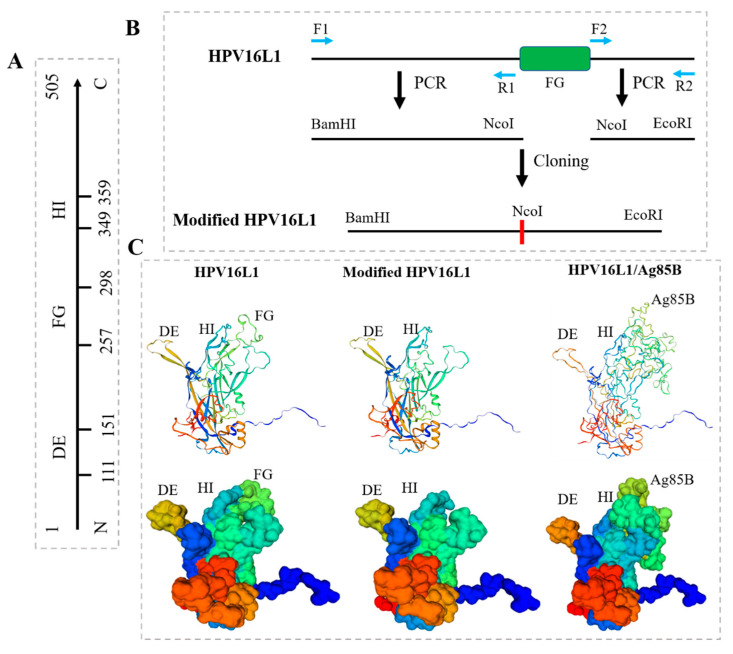
Modification of HPV16L1 and construction of HPV16L1/Ag85B. HPV16L1 contains 505 amino acids in its full length and is characterized by three virus-facing hinge regions between beta folds: DE, FG, and HI loop. (**A**) The sequential construction of HPV16L1. (**B**) Modification of HPV16L1. The sequence on the left side of FG loop was firstly amplified via paired primers, F1 and R1, with the addition of two restriction sites, BamHI and NcoI, respectively. The sequence on the right side of the FG loop was then amplified via paired primers, F2 and R2, designed on the C-terminal of HPV-16L1 and the right side of the FG loop sequence with the addition of two restriction sites, EcoRI and NcoI, respectively. The above left sequence was cloned into pBlu plasmid via BamHI and NcoI, and the right sequence was then inserted into the recombinant pBlu vector carrying the left sequence via NcoI and EcoRI, thus obtaining the HPV-16 L1 gene without the FG loop and forming a restriction site, NcoI. (**C**) Prediction of monomer structures of HPV16 L1 and chimeric HPV16L1/Ag85B by using the SWISS-model server. 7cn2.1.R was chosen as the structural template to build HPV16 L1 (**left**), modified HPV16 (**middle**), and HPV16L1/Ag85B (**right**) homology models. HPV16 L1 was characterized by three loops, DE, FG, and HI, while only DE and HI loops were labeled in the modified HPV16. The position of the FG loop was replaced by Ag85B sequence t in HPV16L1/Ag85B.

**Figure 4 viruses-15-02123-f004:**
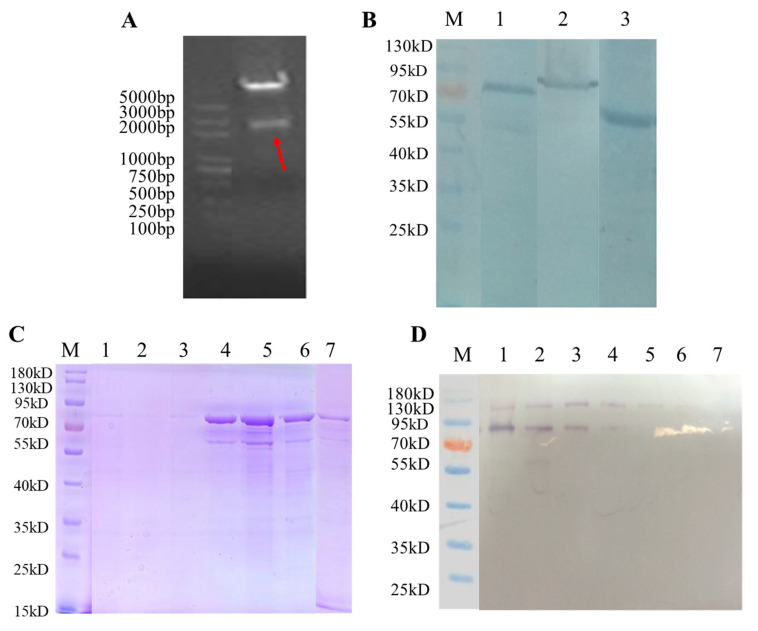
Production of the chimeric HPV16L1/Ag85B VLPs in *Pichia pastoris.* (**A**) The fusion HPV16L1-Ag85B sequence was digested from the recombinant pBlu vector. BamHI and EcoRI restriction enzymes were used to cut the HPV16L1-Ag85B fusion sequence from the recombinant pBlu vector. As the red arrow indicates, the length of the digested fragment was identified via DNA gel electrophoresis. (**B**) Line 1 and 2 indicate that HPV16L1/Ag85B could be recognized by both anti-HPV16 monoclonal antibody and anti-Ag85B polyclonal antibody. Line 3 indicates that HPV16L1 could be recognized by anti-HPV16 monoclonal antibody as a positive control. (**C**,**D**) SDS-PAGE and WB analysis of purified HPV16L1/Ag85B. HPV16L1/Ag85B VLPs were purified via discontinuous Optiprep density gradient centrifugation (27–33–39%) and detected via SDS-PAGE and WB using anti-HPV16L1 antibody.

**Figure 5 viruses-15-02123-f005:**
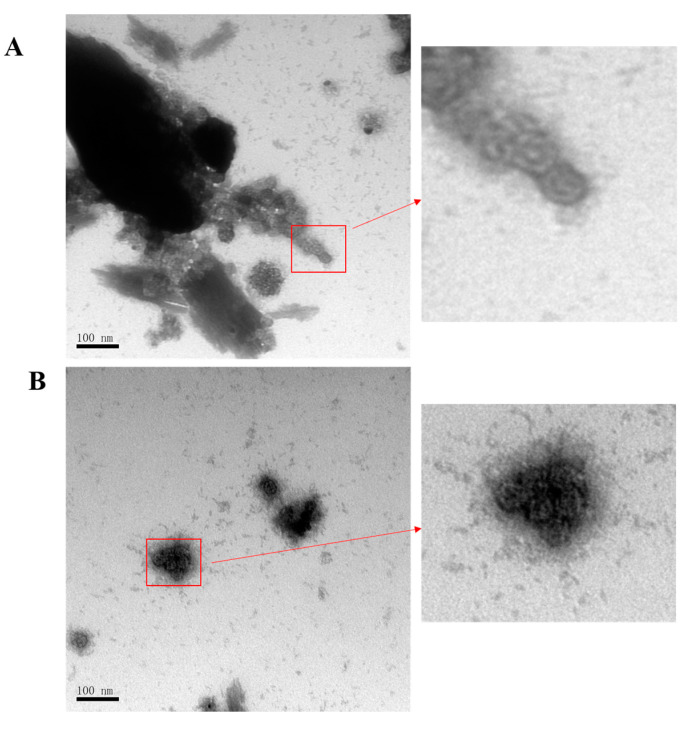
Electron micrographs of HPV16L1/Ag85B VLPs. (**A**) Morphology of the chemic HPV16L1/Ag85B VLP. (**B**) Morphology of HPV16L1 VLP as a positive control. Purified VLPs equilibrated with 20 mM Tris-HCl (pH 7.4; 137 mM NaCl) were layered onto copper TEM grids for 1 min and negatively stained using 2% uranyl acetate solution for 1 min. The grids were placed in a dehumidifier chamber at least 2 h before observation. Images were acquired using a TEM. The bar represents 100 nm at a magnification of 150,000.

**Figure 6 viruses-15-02123-f006:**
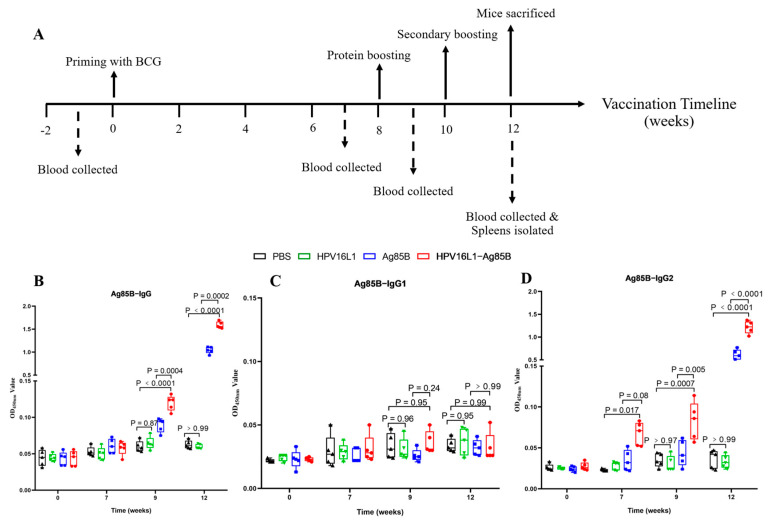
Induction of humoral immune responses by the chemic HPV16L1/Ag85B VLP in mice. (**A**) The immunization schedule. The mice were immunized by priming them with BCG, followed by boosting with vaccine candidates twice at two-week intervals. Blood samples were collected before priming with BCG and after one or two weeks per immunization, and spleen samples were collected two weeks after the final immunization for cytokine analysis. (**B**–**D**) Ag85B-specific IgG, IgG1, and IgG2a in female mice. Serum samples from the different immunized groups at different vaccination time points were collected, and the antibody titers of IgG and two of its subtypes, IgG1 and IgG2a, were detected via an indirect ELISA coated with recombinant Ag85B. Data are shown as median ± S.D. One-way ANOVA test was conducted to compare differences between groups.

**Figure 7 viruses-15-02123-f007:**
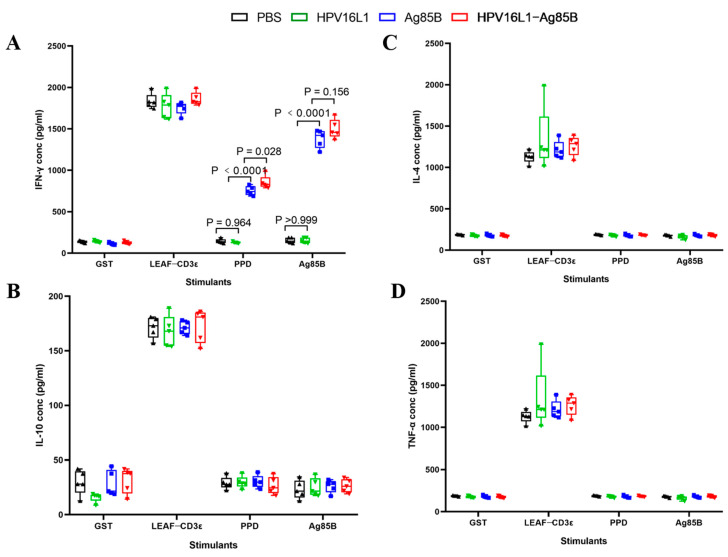
Cytokine production by the splenocytes stimulated with Ag85B. (**A**) Separated splenocytes from the four groups were all stimulated with GST, CD3ε antibody, PPD, and Ag85B, and supernatants were collected after 48 h to be used for IFN-γ quantification. (**B**–**D**) The concentrations of IL-2, IL-10, and TNF -α from above samples were quantified via ELISA. Data are shown as median ± S.D. One-way ANOVA test was conducted to compare differences between groups.

**Table 1 viruses-15-02123-t001:** Primer sequences.

Primer Name	Targeted Gene	Sequence
F1	left side of FG loop	GGGGATCCAAAAAAATGTCTTTGTGGCTTCCATCCGAA GC
R1		CCCCATGGGACAAACATCTGCTCACGTC
F2	right side of FG loop	CCCCATGGCCTACTCCATCCGGTAGTATG
R2		GGGAATTCCTATTACAACTTTCTTTTC
F3	AOX1	GACTGGTTCCAATTGACAAGC
R3		GCAAATGGCATTCTGACATCC

## Data Availability

Not applicable.

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
