# Peer review of "Nano-Sized Chimeric Human Papillomavirus-16 L1 Virus-like Particles Displaying Mycobacterium tuberculosis Antigen Ag85B Enhance Ag85B-Specific Immune Responses in Female C57BL/c Mice"

_viruses, 2023, doi:10.3390/v15102123_

Round 1
Reviewer 1 Report
The paper describes a fusion between the HPV L1 subunit and the TB antigen Ag85. The overall aim is to generate a vaccine that is superior to the live attenuated vaccine BCG. Thus the paper is of potential value and interest. However, the data presented do not reach that aim as there is no comparison to the BCG vaccine.
Comments:
The authors describes in high detail the clinical trials of other candidates in the introduction. As this paper is very early it would give the reader a better understanding to introduce the pre clinical studies of AG85 and how that compares to BCG, and why there is no licensed vaccine with AG85 if this is the key antigen.
The production methods for the vaccine can potentially be adapted to cGMP. However, no information on yields are presented. This would be an important aspect for further development.
The EM pictures demonstrates what appear to be aggregated particles (both constructs). Dynamic light scattering assays would be ideal for assessment of particle composition.
For the immunogenicity part, why was BCG only used as prime and not in a full regimen? This would in my view be the only way to compare whether the current fusion vaccine L1-AG85 is inferior or superior to BCG.
Have the authors considered that most humans have been infected by HPV and/or vaccinated? How would that affect the response towards the fusion vaccine. My guess would be that priming with natural HPV infection or L1 vaccination may generate a T cell response that could limit the T cell response towards the AG85 epitopes.
Small mistakes on grammar throughout the paper.
Reviewer 2 Report
This manuscript reported that chimeric HPV16L1/Ag85B VLP could induce higher Ag85B-antigen-specific antibody responses and displayed significant Ag85B-specific T-cell immune responses in vivo compared to recombinant Ag85B antigen only. The display of Ag85B-antigen on the VLP platform effectively increases its potency. The characterization is well done, but the presentation is subpar. The paper is interesting and the results are nicely explained. It is recommended for publication after the following minor revisions.
1. In most SDS-Page gels the molecular weight and the bands are not at the same level as the results it is hard to figure out which is for what. Please correct this issue.
2. Why are only a few VLPs observed in TEM. In addition, the given TEM, and VLPs are also not clearly visible. A clear TEM picture should be obtained here.
3. In addition to TEM, the author must include the HPV16L1 and HPV16L1/Ag85B VLPs size exclusion chromatography. Because TEM was performed in the dried state. We should be able to see the particles in their native state.
Reviewer 3 Report
Nano-sized chimeric human papillomavirus
Summary: The authors describe establishing a VLP expression system for developing a cost-effective vaccine for TB. The VLP are formed from a chimeric HPV16L1 capsid that has TB Ag85B grafted into its FG loop. The chimeric capsid protein is expressed using the Picia pastoris expression system. The authors anticipate VLP to assemble either during expression or purification. The authors also test for antigen-specific antibody response when mice are injected with the chimeric VLP. The results suggest that VLP injected mice express significantly higher Ag85B-specific IFN-gamma T cell response.
The work is important for improved and cost-effective TB vaccine development, which is much needed in unindustrialized countries. The manuscript needs major revision to address important concerns.
1. There are many grammatical errors in need of correction. These errors may be the reason for my major concerns. Perhaps the authors miscommunicated their ideas/findings. I strongly suggest using the services of a native English speaker to correct these. There are too many to cover in my review.
2. All figure captions must be improved. Important information is lacking.
3. Line 130: The 10nm to 80nm is quite a large discrepancy. Can the authors provide some explanations for why the VLP are so heterogenous?
4. Line 172: “Enough recombinant…” The word Enough is not scientific. Please use concentration/amount.
5. Lin246-247: “… Ag85B was mainly expressed in the body inclusion, but not the culture supernatant.” What is the body inclusion? Is this inclusion bodies? The methods section lacks a protocol for purification of Ag85B. This must be included. Figure 2B and 2C. What are the two bands in each figure? Is the ~26kDa band Ag85B? What is the ~47kDa band? Looking at these images, I do not agree with the reported 90.08% protein purity. These concerns are important because they question the significance of differential antibody expression against injections of Ag85B and HPV16L1/Ag85B.
6. Lines 267-271: Rephrase this section. Clearly communicate that 7CN2 is a PDB entry of …
a. What do the 1.R in 7cn2.1.R represent?
b. What is the PDB entry used for the Ag85B template? A search of the PDB suggests that entry 1F0N is the structure of Ag85B. Is this the template used for the homology modeling?
c. What is the size of the Ag85B insert? Lin158 indicates that it’s from amino acid position 41 – 285 (244 amino acids). A 244 amino acid peptide has a molecular weight of ~26.8kDa. The model for Ag85B in Figure C does not look like a 26.8kDa protein. It looks significantly smaller. Please explain the discrepancy.
7. Line 288: What is meant by “… and the colonies with multiple copies…” Multiple copies of what? How was it determined that these colonies had multiple copies?
8. Figure 4 and caption:
a. Please position the molecular weight (MW) of each standard (marker) lane next to the band. Some MW are below/above the band (in between 2 bands).
b. Figure 4C and 4D. It seems that two different markers are used. If so, why? If not, why do I not see as many bands in Figure 4D lane M as in Figure 4C lane M.
c. Figure 4C and D. Why are there 2 bands in each lane? Please identify each band.
d. If HPV L1 is ~52kDa, and Ag85B is ~27kDa, I would anticipate the fusion proteins to be ~79kDa. Why do I not see a band near ~79kDa in Figures 4C and 4D.
e. The MW of Figure 4B (lanes 1 and 2) agree with the ~79kDa MW, but do not agree with the bands in Figure 4C and 4D.
9. Figure 5A and 5B. I do not see VLP. A more appropriate image would show 30+ VLP in one image to show that assembly is robust and not a rare event. This means that it would be better to take one low magnification image where many VLP are observed, and one high magnification image where the morphology of a few can be observed.
10. Line 311: “ equally assigned into PBS..” This does not make sense.
11. Figure 6 and 7: Do not use bar plots, as these mask underlying data where a few data points may have tremendous effect on p-values. Either use box scatter plots, or violin plots -both show all data points.
12. Line 371: The authors have not reported any experiments where the stability (e.g. melting temperature) or self-assembly of VLP are tested. These would compare the stability and yield of VLP formed by HPV16L1 and HPV16L1/Ag85.
In need of improvement by an English speaker.
